# BRAIN-TO-4D: 4D GENERATION FROM FMRI

## ABSTRACT

Brain-computer interface (BCI) with functional magnetic resonance imaging (fMRI) has enabled new communication interfaces for many real-world applications, *e.g.*, fMRI to image or video. While useful for specific scenarios (*e.g.*, neurofeedback), the existing functions are limited in offering immersive user experience as required by more complex applications (*e.g.*, virtual reality). We thus propose **Brain-to-4D**, a more powerful yet challenging BCI function to construct 4D visuals including both video and 3D directly from brain fMRI signals. In reality, however, it is infeasible to acquire brain signals for multi-view 4D stimuli for training data collection due to the instantaneity nature of brain activities. Typically, brain fMRI data exhibit significantly large variation. To address both obstacles, we introduce **WSf4D**, a novel **W**eakly **S**upervised decomposed **f**MRI-to-**4D** generation approach, characterized by foreground-background decomposition for supervision dividing and fMRI multifaceted vector quantization for noise suppression. To explore the application of the new task Brain-to-4D and our solution WSf4D, we conduct analysis and diagnosis on various brain regions by encoding distinct visual cortex groups. Extensive experiments show that WSf4D can accurately generate multi-view consistent 4D scenes semantically aligned with raw brain signals, indicating meaningful advancements over existing approaches on the potentials of neuroscience and diagnosis.

## 1 INTRODUCTION

Brain-computer interfaces (BCIs) (Saha et al., 2021; Rashid et al., 2020) have been increasingly recognized for their capacity to enable new useful communication means directly through brain activities, underpinning extensive applications in neuroscience (*e.g.*, spatiotemporal functionalities analysis (Yu et al., 2023a; You et al., 2024; Wu et al., 2020)), healthcare, diagnosis, assistive technologies like virtual reality (see Section A.1 for more discussion on applications). As one of the main non-invasive BCI approaches, functional magnetic resonance imaging (fMRI) has been extensively capitalized for implementing various BCI functions. Indeed, with recent advance of generative AI, latest fMRI decoding methods allow to decode a few visual formats such as images (Takagi & Nishimoto, 2023; Lin et al., 2022; Chen et al., 2023b), videos (Wang et al., 2022; Chen et al., 2024a) or 3D shapes (Gao et al., 2023) (see Figure 1 **(a)**). However, that is still largely limited for practical applications as mentioned above due not lacking of immersive communication and interactions.

In this paper we propose for the first time a more powerful BCI function, **Brain-to-4D**, that decodes the brain fMRI signals to 4D visual format encapsulating both video and 3D components (Figure 1**(b)**). This opens new avenues for spatiomotion-related neuro-science and interactive brain health diagnosis (Figure 1**(c)**), providing more dynamic, responsive, and tailored virtual environments. Also, this task gives rise to even bigger challenges. The *first* challenge is *no full supervision*, as acquiring brain signals for 4D stimuli is infeasible in practice (Zhang et al., 2021b) – brain response signals are instantaneous, disabling simultaneous capturing of *multi-view* brain stimuli in reality. The *second* challenge is with *large variation* of brain fMRI due to both intrinsic complexity of brain activities and uncontrollable capturing factors. The interconnected nature of these challenges makes this problem even more difficult. However, inspired by the human ability to continuously perceive dynamic scenes across space and time from fleeting thoughts (Heft, 2010; Kiverstein & Rietveld, 2021; Wang & Spelke, 2002), we are determined to tackle the fMRI-to-4D problem.

Figure 1: **Comparing fMRI signals based BCI functions.** **(a)** Subject to respective visual stimuli, prior fMRI to image, to 3D shape, and to video functions *cannot* support continuous, immersive user experience. **(b)** By generating dynamic 3D scenes directly from fMRI data, our Brain-to-4D enables brain-driven virtual reality, making **(c)** many profound applications such as spatiomotion-related Neuroscientific research and brain health diagnosis possible.

To address the aforementioned challenges, we develop a novel **W**eakly **S**upervised decomposed **f**MRI-to-**4D** generation approach, **WSf4D**, allowing to generate dynamic 3D scenes directly from brain fMRI signals. Our key idea is blending *partial* supervision in correspondence across two modalities – 4D object targets (*i.e.*, foreground) and 3D background in video format. This leads naturally to a scene decomposed architecture: first converting fMRI input into foreground and background representations for respective processing and optimization, then composing them back view by view to the desired 4D visual format with a holistic integrated scene. Critically, this decomposition provides an opportunity of incorporating 2D (partial) supervision available seamlessly. To suppress the signal variation, we compress fMRI signals into discrete semantic vectors so that redundant and noisy information can be filtered out, along with improved computational efficiency in lower dimension space. When applying WSf4D to neuroscience (Figure 1(c)), we encode distinct visual cortex groups, such as full brain regions and V1, to study the function of V1 region. Besides, we add noise to fMRI of V1 to imitate disordered brain for diagnosis.

In summary, we make the following **contributions**: **(i)** To power BCI function with immersive use experience, we introduce a novel, more challenging yet more powerful function, Brain-to-4D, transforming brain fMRI signals to dynamic 3D scenes. **(ii)** We propose a novel weakly supervised decomposed learning method, ***WSf4D***, in a foreground and background decomposed architecture, learnable at the absence of fully supervised fMRI-4D paired training data. **(iii)** For evaluation, we create a new benchmark on top of a previous fMRI-video dataset (Wen et al., 2018) with extended text annotations. We conduct extensive experiments to validate the superior performance of our model over previous alternative in generating dynamic 3D scenes with brain signals.

## 2 RELATED WORK

**Neural decoding for BCIs** Existing BCI functions (Saha et al., 2021; Rashid et al., 2020) are primarily confined to static 2D interactions (Lawhern et al., 2018; Guger et al., 2024; Abdulkader et al., 2015). Previous neural decoding studies (Beliy et al., 2019; Buckner, 1998; Roelfsema et al., 2018) are also limited to 2D images (Beliy et al., 2019; Takagi & Nishimoto, 2023; Chen et al., 2023b; Scotti et al., 2023), videos (Chen et al., 2024a; Lu et al., 2024) and 3D geometry (Gao et al., 2023), making them hard to support continuous, three-dimensional immersive user experience. We thus propose Brain-to-4D function for more seamless and intuitive interaction, providing a significant step forward for practical applications.

**Weakly supervised learning** Previous weakly supervised learning approaches (Zhou, 2018; Mahajan et al., 2018; Zheng et al., 2021) typically focus on incomplete (Settles, 2009; Zhu, 2005; Huang et al., 2010; Chen et al., 2020), coarse (Dietterich et al., 1997; Foulds & Frank, 2010; Wei et al., 2016), or inaccurate supervision (Frénay & Verleysen, 2013) assuming uni-modality labels are available. In contrast, our fMRI-to-4D framework needs to tackle mismatched modality, with 2D video supervision partially corresponding to 4D scene targets. By extracting and integrating information from 2D videos into 4D scenes, our WSf4D expands the scope of weakly supervised learning due to its ability of bridging mismatched modalities.

**3D and 4D generation** Recent advancements in text/image-based 3D generation (Poole et al., 2023; Lin et al., 2023; Wang et al., 2023; Tang et al., 2024; Liu et al., 2023; Shi et al., 2023) are predominantly based on strong 3D representations, including NeRF (Mildenhall et al., 2020), DMTet (Shen et al., 2021) or Gaussian splatting (Kerbl et al., 2023), which leverage score distillation sampling (Poole et al., 2023) (SDS) and extensive 3D datasets (Deitke et al., 2023; Yu et al., 2023b; Wu et al., 2023). With the emergence of 4D representations (Wu et al., 2024; Pumarola et al., 2020; Cao & Johnson, 2023; Yang et al., 2024b; 2023), these techniques have also been extended to generate dynamic 3D scenes (Jiang et al., 2024; Ren et al., 2023; Tang et al., 2024). Our approach takes a step further by integrating rich representations from brain signals as guidance to seamlessly bridge the gap between fMRI and 4D generation, highlighting its superiority in generating immersive and accurate 3D/4D environments from neurological data. An extended discussion can be found in Section A.3 in the supplementary material.

## 3 METHOD

### 3.1 PRELIMINARY

**Deformable 3D Gaussian splatting** 3D Gaussian splatting (3DGS) (Kerbl et al., 2023) represents a 3D scene with a set of Gaussians. Each Gaussian is characterized by position mean $\mu \in \mathbb{R}^3$, covariance matrix $\Sigma \in \mathbb{R}^{3 \times 3}$, color $\mathbf{c} \in \mathbb{R}^3$, and opacity $\alpha \in \mathbb{R}$. The color of each pixel results from the 2D projection of these 3D Gaussians and depth pre-sorted volumetric rendering. In dynamic setting, deformable 3DGS (Wu et al., 2024) uses an additional network $\Phi$ to predict the deformation of $S = \{\mu, \Sigma, \alpha\}$ given timestamp $\tau$: $\tilde{S} = \Phi(S, \tau)$, where $\tilde{S}$ denotes the deformed attributes of $S$. With these deformed attributes, we can render images at different timestamp.

**Score distillation sampling** Score distillation sampling (SDS) provides a method for distilling the knowledge from a pretrained diffusion model $\epsilon_\phi$. Specfically, when an image $I$ is rendered from a scene representation (*e.g.* 3DGS) parameterized by $\theta$, the gradient of SDS loss is calculated as:

$$\nabla_\theta \mathcal{L}_{\text{SDS}}(\phi, I_t) = \mathbb{E}\left[ w(t) \left( \epsilon_\phi(I_t; t, c) - \epsilon \right) \frac{\partial I_t}{\partial \theta} \right], \tag{1}$$

where $I_t$ is the perturbed image with noise $\epsilon$ at time step $t$, and $c$ is the condition (*e.g.* text or image).

**Vector quantization** Vector quantization (VQ) involves mapping continuous input embeddings to discrete codebook entries. Given an input embedding $z_{\text{e}} \in \mathbb{R}^D$, the quantized embedding $z_{\text{q}}$ is determined by selecting the closest codebook vector from a set of codebook entries $\{g_j \in \mathbb{R}^D\}_{j=1}^K$ based on $z_{\text{q}} = g_k$, where $k = \text{argmin}_j \|z_{\text{e}} - g_j\|$.

### 3.2 OVERALL FRAMEWORK OF **WSf4D**

We propose **WSf4D**, a pioneering Weakly Supervised decomposed fMRI-to-4D generation framework, depicted in Figure 2. This framework is designed to tackle the challenge of mismatched modalities between 2D video supervision and 4D scene targets, circumventing the need for paired fMRI-4D data. Central to our approach is the decomposition of scenes into foreground and background, enabling tailored processing to blend partial supervision in correspondence across both foreground and background. Initially, fMRI signals $X$ are encoded into multifaceted components, covering both foreground representations $z_{\text{e,Fg}}, \{I_\tau\}_{\tau=1}^T$ and background representations $z_{\text{e,Bg}}, I_{\text{Bg}}$, with

$$\{z_{\text{e,Fg}}, z_{\text{e,Bg}}, I_{\text{Bg}}\} = \{f_{\text{FVE}}, f_{\text{BVE}}, f_{\text{Bg}}\}(f_{\text{b}}(X)), \{I_\tau\}_{\tau=1}^T = f_{\text{Fg}}(X), \tag{2}$$

as detailed in section 3.3. This encoding is optimized by the 2D videos, allowing the model to effectively learn rich and meaningful representations from the complex fMRI data with limited direct supervision. Subsequently, these representations are then extended into the generation of 3DGS-based 4D scene (section 3.4) which is also decomposed with object foreground and scene background. This decomposition strategy targets to separately exploit different multifaceted representations based on their respective characteristics. The foreground involves generating a 4D object using deformable 3DGS (Wu et al., 2024) driven by $z_{\text{e,Fg}}$ and $\{I_\tau\}_{\tau=1}^T$. Concurrently, the background component utilizes spherical 3D Gaussians as representation optimized through $z_{\text{e,Bg}}$ and $I_{\text{Bg}}$. Both components

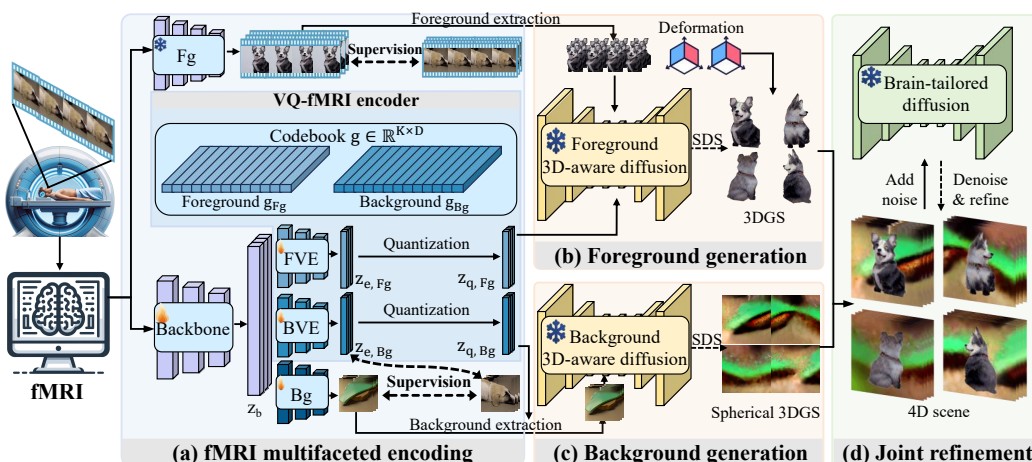

Figure 2: **Overview of our WSf4D.** Without fully supervised fMRI-to-4D training data, our method takes a weakly supervised learning strategy. We start with **(a)** fMRI multifaceted encoding, which includes foreground and background VQ encoders (FVE and BVE), as well as foreground object (Fg) and background scene (Bg) encoders. These encoders can be supervised with 2D videos for extracting meaningful representations from the fMRI. We further model concurrently **(b)** foreground generation over time with deformable 3D Gaussian splatting (3DGS), and **(c)** background generation with spherical 3DGS. **(d)** Finally, we re-composite the foreground and background view by view for allowing joint refinement and optimization using a brain-tailored diffusion model.

are then composed and refined under the guidance of a brain-tailored diffusion to ensure coherence with the original fMRI. This partial supervision with foreground and background decomposition enables us to exploit the highly variable fMRI into realistic 4D scenes when fMRI-4D pairs are impractical to obtain.

## 3.3 VECTOR QUANTIZED FMRI (VQ-FMRI) ENCODING

In pursuit of robust fMRI extraction under sparse training samples, we propose the vector quantized fMRI (VQ-fMRI) encoders to map fMRI data $X$ onto discrete latent space. Specifically, a backbone encoder $f_b$, processes the fMRI data to produce an shared representation $z_b = f_b(X)$. This is then split into foreground and background VQ encoders (FVE and BVE):

$$z_{e,Fg} = f_{FVE}(z_b), z_{e,Bg} = f_{BVE}(z_b), \tag{3}$$

resulting in quantized foreground and background latent space representations:

$$g_{Fg} \in \mathbb{R}^{K_{Fg} \times D_{Fg}}, g_{Bg} \in \mathbb{R}^{K_{Bg} \times D_{Bg}}, \tag{4}$$

where $K_{Fg}$ and $K_{Bg}$ denote the size of latent vectors, and $D_{Fg}$ and $D_{Bg}$ represent their dimensionality. Our designed vector quantization is performed as follows:

$$z_{q,Fg} = g_{k,Fg}, \text{ where } k = \text{argmin}_j \left\| z_{e,Fg} - g_{j,Fg} \right\|, \tag{5}$$

$$z_{q,Bg} = g_{k,Bg}, \text{ where } k = \text{argmin}_j \left\| z_{e,Bg} - g_{j,Bg} \right\|. \tag{6}$$

The quantized foreground embedding, $z_{q,Fg}$, provides semantic and geometric guidance for the foreground reference video generated as $\{I_\tau\}_{\tau=1}^T = f_{Fg}(X)$. The quantized background embedding $z_{q,Bg}$ supports inpainting for the background reference image decoded as $I_{Bg} = f_{Bg}(z_b)$. For further implementation details, refer to section A.2.

One key advantage is its ability to bypass the curse of dimensionality. By constraining latent space size $K \ll n$, we significantly improve model regularization and avoid overfitting (Peng et al., 2023) in high-dimensional feature spaces. Furthermore, our approach significantly reduces KL divergence between empirical and ground truth distributions, as indicated by theorem 3.1. It shows that the quantized latent space $z_q$ yields a much tighter approximation to the true distribution compared to the non-quantized embeddings $z_e$, which is crucial for robust latent representations.

**Theorem 3.1.** *Denote $p(z_e)$ as distribution of the embeddings without vector quantization and $p(\hat{z}_e)$ as the smooth-approximated empirical distribution from samples. Denote $p(z_q)$ and $p(\hat{z}_q)$ as their vector quantized counterparts. Then,*

$$KL(p(z_q)||p(\hat{z}_q)) \ll KL(p(z_e)||p(\hat{z}_e)). \tag{7}$$

Additionally, theorem 3.2 shows our vector quantized approach also significantly reduces entropy. This ensures that the model is less likely to capture irrelevant data-specific noise, thereby enhancing generalization to unseen data.

**Theorem 3.2.** *Denote $L$ as the CLIP (Radford et al., 2021) space boundary size, $H(z_q)$ as the entropy of distribution of vector quantized embeddings, and $H(z_e)$ as the entropy of Riemann-Discrete approximated distribution without vector quantization. Then we have $H(z_e) > H(z_q)$,*

$$H(z_e) - H(z_q) = O\left(\log\left(\frac{L^d}{K}\right)\right). \tag{8}$$

Detailed proof could be found in section A.4 and section A.5 in supplementary material.

### 3.4 FOREGROUND-BACKGROUND DECOMPOSING FOR 4D SCENE GENERATION

Modeling 4D scenes face two challenges: (1) Foreground and background present intrinsically different characteristics (*e.g.*, dynamic vs. static); (2) Camera perspectives in 4D scenes often blur out nearby objects dynamically. To tackle these, we propose decoupling the foreground and background elements of a scene.

**Foreground generation** The foreground is represented by deformable 3D Gaussians, optimized in two stages: static and dynamic (Ren et al., 2023; Yin et al., 2023), driven by foreground video $\{I_\tau\}_{\tau=1}^T$ and the quantized embedding $z_{q,Fg}$. In both stages, 3D Gaussians and its deformation are guided by object-level diffusion models under SDS. Along with mean squared error (MSE) loss under reference views with $I_{ref} \in \{I_\tau\}_{\tau=1}^T$, the total loss $\mathcal{L}_f$ for foreground modeling can be expressed by:

$$\mathcal{L}_f = \lambda_{img}\mathcal{L}_{SDS,img} + \lambda_{text}\mathcal{L}_{SDS,text} + \lambda_{ref}\|\hat{I}_{ref} - I_{ref}\|_2^2, \tag{9}$$

where $\lambda_*$ are balancing weights, with img and text referring to AI (2023) and Shi et al. (2023) guidance, respectively. Furthermore, at static stage we set the first frame $I_1$ as reference image and froze the deformation network $\Phi$ during training. In contrast, the dynamic stage utilizes all the frames, allowing $\Phi$ to be trainable to accommodate temporal variations. Considering the unstable training of Gaussians in the generative manner, we follow Pan et al. (2024a) to manually clip the gradient of rendered image pixel-wisely. This operation significantly reduces the variance of gradients, avoiding intricate densification parameter tuning and leading to improved shape and texture.

**Background generation** The background is represented by 3D Gaussians around a sphere without deformation. A scene-level 3D-aware diffusion model serves as a 2D prior to extend the background image $I_{Bg}$ into a complete 360° environment. The total loss $\mathcal{L}_b$ for background modeling is:

$$\mathcal{L}_b = \lambda_{Bg}\mathcal{L}_{SDS,Bg} + \lambda_{ref}\|\hat{I}_{Bg} - I_{Bg}\|_2^2, \tag{10}$$

where $\lambda_*$ denotes balancing weights and $\mathcal{L}_{SDS,Bg}$ represents SDS under scene-level diffusion.

**Joint refinement** To ensure a cohesive integration of foreground and background, we design a joint refinement stage while maintaining each Gaussian representation. To get the composite image $I_c$, we render both foreground image $I_f$ and background image $I_b$ with a foreground mask $M_f$, and then blend them by:

$$I_c = I_f \odot M_f + I_b \odot (1 - M_f). \tag{11}$$

Then we can further render a composite video $\{I_{c_k}\}_{k=1}^T$ under any viewpoint. At this stage, we introduce brain-tailored diffusion to directly denoise the noise-perturbed video, providing a refined image $I_{refine_k}$ for each frame as supervision. An MSE loss (12) is applied to refine both 4D Gaussians and spherical 3D Gaussians.

$$\mathcal{L}_{refine} = \sum_k \|I_{c_k} - I_{refine_k}\|_2^2 + \|\hat{I}_{ref} - I_{ref}\|_2^2. \tag{12}$$

Figure 3: **ROI (region of interest) interpretability and diagnosis.** Our proposed WSf4D can separately encode distinct visual cortex groups for Neuroscientific research, and could conduct diagnosis on various brain regions.

## 3.5 APPLICATIONS: NEUROSCIENCE INTERPRETABILITY AND DIAGNOSIS

We apply WSf4D to two key applications: neuroscience interpretability and diagnosis (Figure 3). Our design focuses on four specific groups within the visual cortex: primary (V1), associative (V2, V3, V4), dynamic (MT, MST, LIP), and synthesis (TPOJ) visual cortex. For each group, we examine their role by encoding each region of interest (ROI) group separately. To simulate disorder diagnosis, we introduce perturbations to each group and analyze the resulting 4D scenes to evaluate their functional impact.

## 4 EXPERIMENTS

### 4.1 BENCHMARK

**Dataset** Our research extends publicly available fMRI-video dataset (Wen et al., 2018). The fMRI are acquired using a 3T MRI scanner at a repetition time (TR) of 2 seconds, comprising 18 segments of 8-minute video clips, resulting in 4,320 training video-fMRI pairs, and 5 segments for 1,200 testing samples. For each video-fMRI pair, a single frame is randomly selected as the ground truth image for background supervision. Besides, we annotated the video-fMRI samples with foreground objects (Krizhevsky et al., 2009) and background scenes (Bansal, 2019). Lacking 4D annotations, we employ semantic embeddings of these labels as a codebook to supervise our VQ-fMRI encoders.

**Metrics** In line with (Chen et al., 2024a), we employ the Structural Similarity Index Measure (SSIM) for pixel-level accuracy and classification-based score for semantic accuracy with respect to ground truth visual stimuli. The classification score compares the top-1 accuracy between the ground truth and rendered frames across selected $N = 2$ and $N = 50$ classes, with 100 repetition for an average success rate and standard deviation. Both image and video classifiers are used, designed as ICS-$N$ and VCS-$N$, respectively. Additionally, following Yin et al. (2023); Pan et al. (2024b), we incorporate CLIP-T as a 4D metric, which evaluates the temporal smoothness by computing the CLIP similarity between adjacent frames in a rendered video. Except for reporting CLIP-T of videos at specific views in Yin et al. (2023); Pan et al. (2024b), we also adopt a 360° video around the 4D scene which represents the spatial geometry, resulting in CLIP-T-G. For 4D benchmark, we render a 4D model from the front view (reference view), side views and back view, with each view evaluated separately across 100 cases. The SSIM is only applicable to the reference view because there is no ground truth for other views.

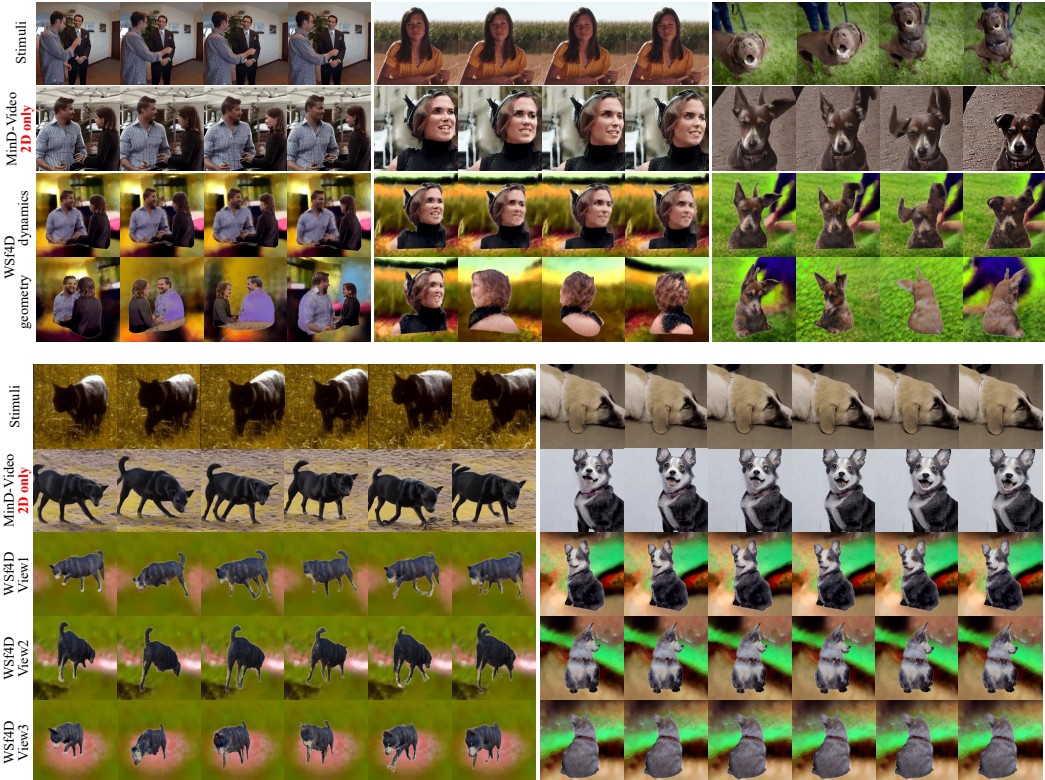

Figure 4: **Multi-view 4D scenarios of WSf4D**. Previous methods (MinD-Video (Chen et al., 2024a)) are limited in **2D** with only 2D supervision. In comparison, WSf4D pinoeers the **Brain-to-4D** function through a novel weakly supervised framework. See the video in supplementary for dynamic results.

Table 1: **Quantitative evaluation.** The pixel-level SSIM score (Wang et al., 2004) is only reported for the front view which is aligned with reference frames. The results of MinD-Video (Chen et al., 2024a) only serve as the reference for front view as it lacks 3D geometry.

| Metrics | MinD-Video front view only | WSf4D front view | side view | back view | mean |
|---|---|---|---|---|---|
| VCS-2 ↑ | $0.9226_{\pm 0.019}$ | $0.9080_{\pm 0.016}$ | $0.8778_{\pm 0.024}$ | $0.8823_{\pm 0.022}$ | $\mathbf{0.8894}_{\pm 0.021}$ |
| VCS-50 ↑ | $0.3602_{\pm 0.022}$ | $0.4135_{\pm 0.020}$ | $0.2607_{\pm 0.017}$ | $0.3303_{\pm 0.021}$ | $\mathbf{0.3348}_{\pm 0.019}$ |
| ICS-2 ↑ | $0.8830_{\pm 0.021}$ | $0.9030_{\pm 0.021}$ | $0.7975_{\pm 0.031}$ | $0.8349_{\pm 0.030}$ | $\mathbf{0.8451}_{\pm 0.027}$ |
| ICS-50 ↑ | $0.3291_{\pm 0.022}$ | $0.2935_{\pm 0.021}$ | $0.1102_{\pm 0.013}$ | $0.1239_{\pm 0.012}$ | $\mathbf{0.1759}_{\pm 0.015}$ |
| SSIM ↑ | 0.2005 | 0.2131 | - | - | **0.2131** |
| CLIP-T ↑ | 0.9434 | 0.9482 | 0.9644 | 0.9622 | **0.9583** |
| CLIP-T-G ↑ | - | - | - | - | **0.9441** |

## 4.2 IMPLEMENTATION DETAILS

Our designed backbone $f_b$, foreground VQ encoder $f_{FVE}$, background VQ encoder $f_{BVE}$ and background scene encoders $f_{Bg}$ are all MLP structures. The foreground object encoder $f_{Fg}$ leverages a pretrained Chen et al. (2024a). Our foreground 3D-aware diffusion use pretrained models from AI (2023) and Shi et al. (2023), while background 3D-aware diffusion employs Sargent et al. (2023). The Brain-tailored diffusion exploit structures from Chen et al. (2024a). More details can be referred in section A.2.

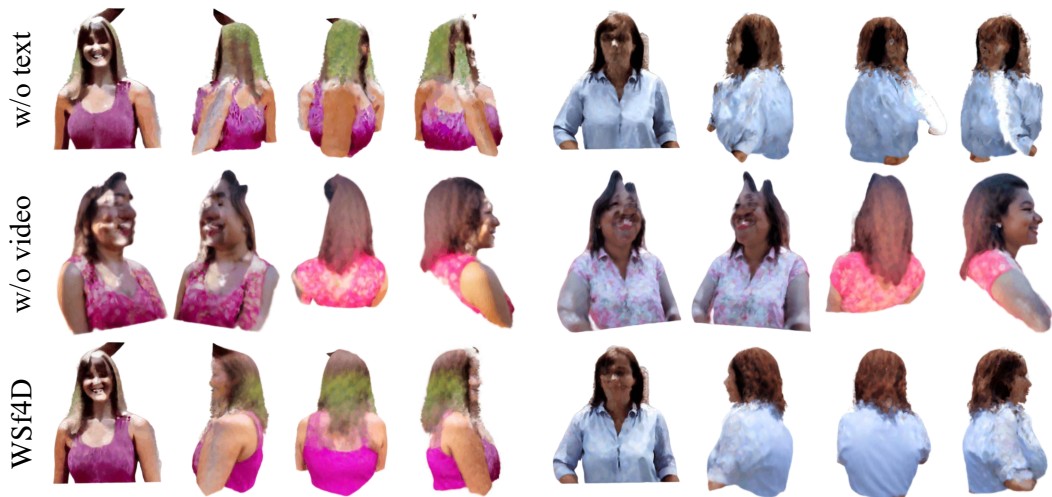

Figure 5: **Ablation on the input of foreground modeling.** Without either text embedding or video frame embedding for 3D appearance guidance, the rendering quality decreases significantly.

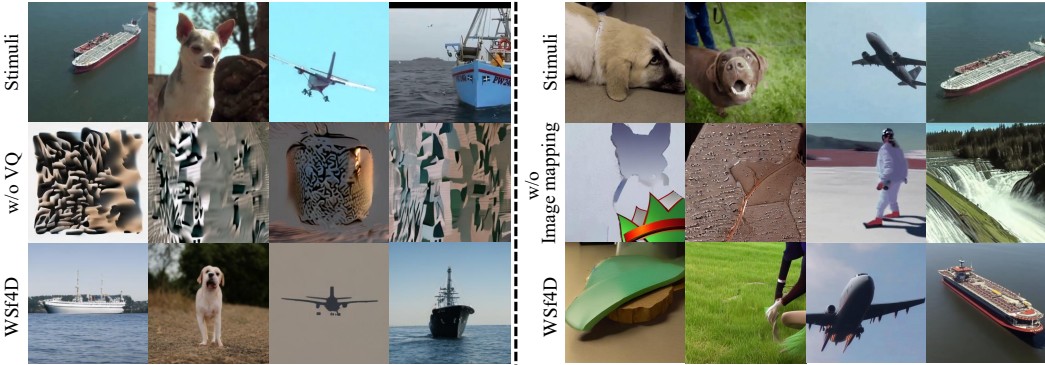

Figure 6: **Left:** Vector quantization (VQ) ablation. Without VQ, the generated images from mapped embedding are totally corrupted. **Right:** Background input ablation. The naive approach of using segmented image from MinD-Video (Chen et al., 2024a) fails to provide mind-related background.

### 4.3 4D GENERATION RESULTS

We present our 4D generation results in Figure 4 and Table 1, which also includes comparisons with MinD-Video (Chen et al., 2024a). For visual results in Figure 4, while MinD-Video is limited to single-view videos, our method extends videos into dynamic scenes with full 3D geometry. Besides, our background branch enables 3D rendering with closer semantic alignment with respect to visual stimuli, such as accurate lakeside scenery and building layout in Figure 12. Our method achieves a higher SSIM score (Wang et al., 2004) from the reference view (front view) as detailed in Table 1. Regarding semantic-level metrics, our method achieves comparable success rates from the reference front view, with slight declines from other views possibly due to the absence of visual stimuli in these views. However, all success rates significantly surpasses the base chance level (2-way: 0.5, 50-way: 0.02). For CLIP-T scores assessing the 4D effect, our results demonstrate both dynamic and spatial smoothness, all outperforming MinD-Video, which focuses on single-view output. Please refer to section A.7 for more visualization results.

### 4.4 ABLATIONS

**Vector quantization** Figure 6 (left) highlights the crucial role of vector quantization (VQ) in fMRI encoding. Without VQ, the MLP embeddings $z_e = f_e(X)$ result in ineffective image generation, which has cosine similarity of only 0.073, caused by high variation with fMRI and data scarcity.

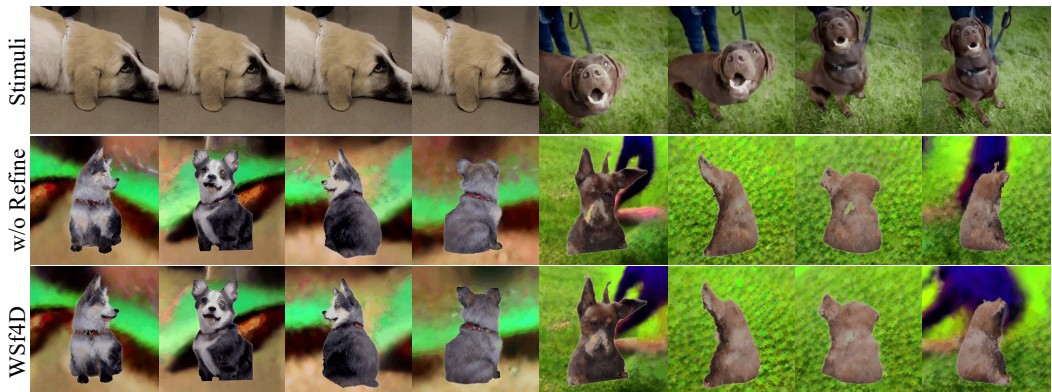

Figure 7: **Ablation for refinement stage** which leads to superior details.

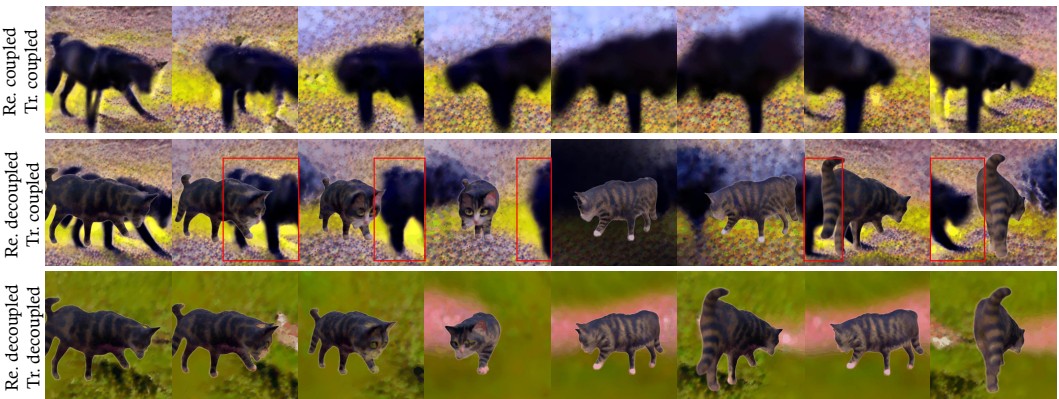

Figure 8: **Ablation on decoupling-coupling.** "Re." denotes representation and "Tr." denotes training. The coupling of representations leads to bad geometry and coupling of training leads to ambiguity.

In comparison, our VQ-fMRI encoder captures the semantic information, with an increased cosine similarity of 0.789, facilitating accurate reproduction of 4D scenes.

**Background extraction**  We ablate the input of background modeling in the right of Figure 6. The baseline method "w/o Image mapping" directly segments the first frame of the video generated by Mind-Video (Chen et al., 2024a) and uses background text embedding for inpainting. This approach often results in images with meaningless content or a mismatch with the ground truth visual stimuli.

**Ablations on decoupled training strategy**  In figure 8, we conduct the ablation study on the decoupled training strategy. We find that the coupling of foreground and background poses the challenge to the optimization of 4D scene, while the decomposition design introduced in section 3.4 achieves the best geometry and avoids the ambiguity between the foreground and background.

**Usage of embeddings**  We further investigate the impacts of text or image embeddings on foreground generation, as shown in Figure 5. Since the reference frames are typically out of distribution of the training data (Deitke et al., 2023) used for 3D-aware diffusion models, the baseline "w/o text" that relies solely on Zero123 guidance fails to produce satisfactory 3D shapes. In addition, the results using only text embedding with MVDream guidance ("w/o video") do not accurately reflect the brain-related images.

**Effect of refinement**  As illustrated in Figure 7, the refinement stage improves the details and eliminates some errors, such as incorrect lighting on the dog's nose and the notch on its back.

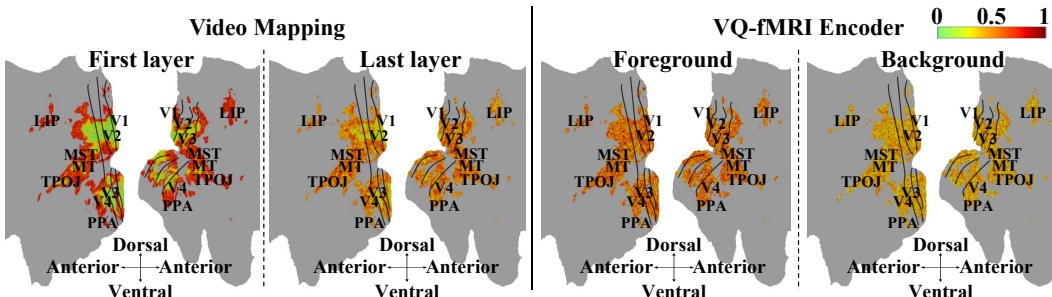

Figure 9: **Voxel-wise importance maps of subject 1.** Early layers of the video mapping concentrate on structural details of brain regions, while deeper layers and the VQ-fMRI encoder increasingly focus on abstract features. Foreground encoding shows significantly more activity than the background.

### 4.5 COMPREHENSIVE ROI ANALYSIS

**ROI importance mapping** We first analyze brain-related mechanisms by visualizing attention maps in the video mapping and encoder weight distributions in VQ-fMRI encoder. As shown in Figure 9, consistent with MinD-Video (Chen et al., 2024a), early video mapping layers prioritize structural aspects of input data, highlighting a clear segmentation of brain regions. High-level visual cortex areas (MT, MST and TPOJ) receive more attention than low-level visual cortex (V1, V2 and V3), reflecting a focus on complex feature extraction. As processing deepens, attention becomes more dispersed, shifting towards holistic and abstract visual features. In contrast, VQ-fMRI encoder demonstrates greater homogeneity among regions, indicating a more holistic visual features. Specifically, the foreground VQ-fMRI encoder identifies more high-value regions than the background encoder, which hints more brain areas are focused on forground object instead of background scenes. Most values in background VQ-fMRI encoder shows a small weight value, indicating their little contribution to background encoding.

**ROI interpretation** The function of each specific ROI group is also analyzed separately (Figure 3(b)). The V1 visual region maintains initial processing of edges, orientations, and spatial frequencies of the scene, confirming its essential role in basic visual feature detection. The associative (V2, V3, V4) cannot independently decode visuals, indicating their reliance on V1 for information processing. Meanwhile, the spatiomotion (MT, MST, LIP) regions could only generate motion and flow, contributing little to complex patterns and shapes. The TPOJ region includes a cohesive visual experience, illustrating its role in information integration. These findings align well with previous research on region-of-interest (ROI) functionality in visual perception (Tong, 2003; Kim et al., 2020).

**ROI diagnosis** These ROI functions points to the potential for ROI diagnsis. As depicted in Figure 3(c), the disorder in either primary (V1) visual regions or associative (V2, V3, V4) regions lead to impairments in overall visual comprehension, supporting the centrality of these regions in foundational and complex visual processing. Disorders in the synthesis (TPOJ) region result in a more comprehensive disruption of scene perception, suggesting its crucial role in integrating visual inputs into a coherent whole. In contrast, the disorder in spatiomotion (MT, MST, LIP) produce only marginal effects, showing their little impact on features and edges.

## 5 CONCLUSION

In this study, we introduce WSf4D, a pioneering framework tailored for the newly proposed Brain-to-4D BCI function, enabling the generation of dynamic 3D scenes from brain fMRI signals for immersive user experience. Through meticulous design, the WSf4D framework overcomes the challenges posed by the absence of fully supervised 4D brain training data and high variation with brain fMRI signals. Our core idea is to adopt a weakly supervised learning approach that streamlines weak, partial supervision from the pre-existing fMRI-video and single-view-to-3D in a background and foreground decoupled architecture. Experimental results have demonstrated the capability of WSf4D in decoding time-continuous and view-consistent 4D visuals closely aligned with the underlying brain activity. We hope this work can open up and foster more advanced research and applications in BCI and neuroscience studies.

## 6 ETHICS STATEMENT

We believe that our proposed task and method has promising applications in Brain-Computer Interfaces. However, every method that learns from data carries the risk of introducing biases. In the fMRI encoding stage, all the encoders are trained on open-source brain datasets described in Section 4. The subsequent generation stage is based on the open-source diffusion models that are pre-trained on the data from the Internet. Therefore, work that bases itself on our method should carefully consider the consequences of any potential underlying risks and biases.

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

## A    SUPPLEMENTARY MATERIAL

### A.1    LIMITATION AND FUTURE WORK

As a preliminary exploration of Brain-to-4D function, our proposed weakly supervised framework is highly open and integratible, able to continuously and readily benefit from any improvement of any components involved. The overall quality of the generated 4D content is currently constrained by fMRI decoding (Chen et al., 2024a) and generation models (AI, 2023; Sargent et al., 2023; Shi et al., 2023). Furthermore, our method occasionally generates blurry outputs. We believe that above problems will eventually be addressed with developments of neural decoding (e.g. incorporation of (Lu et al., 2024)) and 4D reconstruction.

Our application on spatiomotion-related neuro-science and interactive brain health diagnosis also could be further developed with improved models and clinical experiments. The other potential real-world applications for WSf4D include:

(1) Brain-driven virtual reality for immersive communication and interaction, such as enabling users to navigate virtual spaces using only their thoughts. Advanced gaming experiences controlled by brain signals can offer new levels of immersion and interaction.

(2) In neurorehabilitation, it can simulate realistic environments for stroke patients to practice daily activities.

(3) Brain-driven creativity allows artists to produce 3D movies and artistic expressions using only their thoughts, thus unlocking new forms of immersive artistic expression.

(4) Educational tools can provide interactive, brain-responsive simulations, such as virtual science experiments controlled by students' brain activity.

## A.2 IMPLEMENTATION DETAILS

**Encoding**  In the VQ-fMRI encoder, the backbone $f_b$ first employs an MLP to map fMRI data into a 4096-dimensional vector. This is followed by four MLPs with residual connections to further extract fMRI features. The output is then transformed into $257 \times 768$-dimensional shared feature representation $z_b$. Both the foreground VQ encoder (FVE) and the background VQ encoder (BVE) use two-layer MLPs to map this shared feature representation into the VQ-embedding space $z_{q,obj}, z_{q,env}$. The codebook dimensions for foreground modeling are set to $D = 77 \times 1024$, aligned with Shi et al. (2023), while the background modeling follows Takagi & Nishimoto (2023) with dimensions of $D = 77 \times 768$. Given the practical challenges in acquiring sufficient 4D stimuli for end-to-end optimization, these codebooks are crafted around specific categories of foreground objects Krizhevsky et al. (2009) and background scenes Bansal (2019).

For foreground modeling, a model in Chen et al. (2024a) is used to map fMRI data to a reference video that guides appearance and dynamics. We then segment each frame of the video to extract the foreground with total $T$ frames, which are denoted by $\{I_\tau\}_{\tau=1}^T$. Typically, the video content includes cropped scenes or real people, which diverges from the distribution of existing 3D datasets. To bridge this gap, the VQ-fMRI encoder maps fMRI into an text embedding $z_{q,obj}$ for better semantic and geometric guidance. Background encoding starts with generating an background reference image from fMRI. An intuitive approach involves reusing segmented images from video branch in foreground encoding, but this method faced two drawbacks: (1) these frames predominantly feature foreground elements, restricting accessible background information and (2) the backgrounds are not consistent across different frames. To overcome these challenges, we generate this background image directly from the shared representation, and the image is optionally inpainted using scene-level text embedding $z_{q,env}$ from VQ-fMRI encoder. Training all fMRI encoders is a one-time process that takes approximately two days on one NVIDIA A6000 GPU. Once completed, the parameters are fixed for subsequent 4D generation from any fMRI.

**Generation**  In generation stage, we implement our pipeline based on the DreamGaussian4D (Ren et al., 2023), a framework focusing on efficient 4D generation. Training involves 500 steps for static foreground and background, 1,000 steps for dynamic foreground, and 50 steps for joint refinement. The Gaussians are initialized with 5,000 random points for foreground inside a sphere of and 200,000 random points for background around a sphere of radius 5. Densification is performed every 50 steps. For balancing weights, we set $\lambda_{img} = 1, \lambda_{text} = 0.5, \lambda_{ref} = 10,000, \lambda_{env} = 1$. For diffusion guidance, we use pretrained models from Stable Zero123 (AI, 2023) and MVDream (Shi et al., 2023) object-level 3D-aware diffusion, use adopt ZeroNVS (Sargent et al., 2023) as 2D prior in scene-level 3D-aware diffusion, and apply MinD-Video (Chen et al., 2024a) for Brain-tailored diffusion. The whole generation pipeline takes about 30 minutes on one NVIDIA A6000 GPU. Following this, the parameters for 4D Gaussian splatting are saved, enabling future inference processes. This setup allows for an inference speed of 15 frames per second (FPS), supporting real-time interaction.

## A.3 RELATED WORK

**Neural decoding for BCIs**  BCIs aim to establish communication links between the brain and computers or other external devices (Saha et al., 2021; Rashid et al., 2020; Kawala-Sterniuk et al., 2021; Wolpaw et al., 2002; Nicolas-Alonso & Gomez-Gil, 2012). However, BCI research is primarily confined to static 2D interactions (Lawhern et al., 2018; Guger et al., 2024; Abdulkader et al., 2015; Zander & Kothe, 2011) which do not support continuous, three-dimensional immersive experiences. Existing neural decoding studies have focused on extracting essential representations (Buckner, 1998; Roelfsema et al., 2018) of brain signals for tasks like visual content decoding (Naselaris et al., 2011; Kamitani & Tong, 2005; Haxby et al., 2001; Haynes & Rees, 2005; Thirion et al., 2006; Georgieva et al., 2009) and object recognition (Wen et al., 2018; Horikawa & Kamitani, 2017; Groen et al., 2018). However, they often struggle to create detailed visuals directly from brain signals. These investigations have also facilitated advancements in reconstructing images (Beliy et al., 2019; Li et al., 2024), videos (Wang et al., 2022; Chen et al., 2024a; Lu et al., 2024) and geometry (Gao et al., 2023; Yang et al., 2024a; Gao et al., 2024) from fMRI data using techniques such as generative adversarial networks (Schoenmakers et al., 2013; VanRullen & Reddy, 2019; Shen et al., 2019; Dado et al., 2022; Seeliger et al., 2018; Gu et al., 2022; Ozcelik et al., 2022) and latent-space diffusion (Takagi & Nishimoto, 2023; Lin et al., 2022; Ozcelik & VanRullen, 2023; Chen et al., 2023b; Scotti et al., 2023;

Gao et al., 2023). Restricted by high cost of large-scale brain stimuli containing both multi views and time continuity, all these reconstructions are limited to single view or static objects, which pose severe limitation on immersive user experience under BCIs. WSf4D advances beyond these achievements by offering more seamless and intuitive interaction that leverage both spatial and temporal dimension interactions, providing a significant step forward in the practical application of BCIs.

**Weakly supervised learning**  Weakly supervised learning targets at situation with insufficient training dataset (Zhou, 2018; Mahajan et al., 2018; Zheng et al., 2021). Previous approaches typically focus on three key situations: incomplete supervision with mostly unlabelled data (Settles, 2009; Zhu, 2005; Huang et al., 2010; Chen et al., 2020), inexact supervision with only coarse-grained labels (Dietterich et al., 1997; Foulds & Frank, 2010; Wei et al., 2016), and inaccurate supervision with partially incorrect labels (Frénay & Verleysen, 2013). These methods are effective in tasks like object detection (Zhang et al., 2021a; 2018; Tang et al., 2018; Yang et al., 2019; Nag et al., 2022), localization (Choe & Shim, 2019; Jiang et al., 2019; Hou et al., 2018), and segmentation (Zhang et al., 2020; Ahn et al., 2019), where similar modality labels are available. In comparison, our fMRI-to-4D task face a novel challenge of mismatched modality supervision, where the available 2D video labels only partially correspond to the target 4D scenes. Our WSf4D fill this modality gap by squeezing available information from available 2D videos, and then distilling and integrating this information into 4D scenes. This pushes the boundaries of weakly supervised learning by advancing weakly supervision across mismatched modalities.

**3D and 4D generation**  Recent advancements in 3D and 4D content generation have predominantly utilized inputs such as text, images, and videos. The core of these innovations stems from techniques like score distillation sampling (Poole et al., 2023) (SDS) and the exploitation of extensive 3D datasets (Deitke et al., 2023; Yu et al., 2023b; Wu et al., 2023). At the object-level, numerous works (Poole et al., 2023; Lin et al., 2023; Chen et al., 2023a; Wang et al., 2023; Tang et al., 2024; Yi et al., 2024) employ SDS to train fundamental 3D representations, including NeRF (Mildenhall et al., 2020), DMTet (Shen et al., 2021) or Gaussian splatting (Kerbl et al., 2023). Following research continues into training 3D-aware diffusion models for improved geometric consistency (Liu et al., 2023; Shi et al., 2023; Liu et al., 2024; Voleti et al., 2024; Chen et al., 2024b). With the development of fundamental 4D representations (Wu et al., 2024; Pumarola et al., 2020; Cao & Johnson, 2023; Yang et al., 2024b; 2023), the extension for 4D generation fields have been explored. For example, Consistent4D (Jiang et al., 2024) proposes video-to-4D task through a tailored dynamic NeRF with SDS. DreamGaussian4D (Ren et al., 2023) extends the 4D function of DreamGaussian (Tang et al., 2024) to further reduce optimization time with Gaussian splatting. However, these methods often struggle with in-the-wild scenes. DreamFusion (Poole et al., 2023) attempts to model the background using a small coordinate multi-layer perceptron (MLP) distilled by a text-to-image diffusion model, which leads to blurry results. Previous efforts (Yu et al., 2021; Jain et al., 2021) have aimed at single-image novel view synthesis but are confined to a limited range of camera viewpoints. ZeroNVS (Sargent et al., 2023) employs a scene-level diffusion model for novel view synthesis. In comparison, WSf4D not only leverages this prior but also innovates further by optimizing a Gaussian sphere for background modeling. Moreover, WSf4D takes a step further by integrating brain signals as inputs and designing an efficient fRMI encoder to seamlessly bridge the gap between brain and various diffusion models, underscoring its superiority in generating immersive and accurate 3D/4D environments from neurological data.

### A.4    Proof of Theorem 3.1

In sparse sampling where the dimensionality of the encoded latent space $d = \dim(z_\mathrm{e})$ significantly exceeds the number of training samples $n$, that is $d \gg n$, the probability distribution $p(z_\mathrm{e})$ is not adequately represented. The empirical distribution $p(\hat{z}_\mathrm{e})$, which is approximated from a limited number of samples, fails to capture substantial portions of the probability mass inherent to $p(z_\mathrm{e})$.

For any $\delta > 0$, we consider a smooth-approximated empirical distribution encompassing a neighborhood with radius $r$: let $\hat{z}_\mathrm{e}$ be points in the encoded space such that $\|\hat{z}_\mathrm{e} - t_i\| > r$ for all $i \in \{1, \dots, n\}$ with $t_i$ representing the training samples. For these points, it holds that $0 < p(\hat{z}_\mathrm{e}) < \delta$.

Denote $R_i$ as the union of all proximal areas around the training samples:

$$R_i = \bigcup_{i=1}^{n} U_i, \quad \text{where } U_i = \{u \in A : \|u - t_i\| \le r\}, \tag{13}$$

and let $R_o$ represent the complement region in the latent space $A$, which is far from the training samples:

$$R_o = A \setminus R_i. \tag{14}$$

Then the KL divergence without Vector Quantization will become:

$$KL(p(z_\mathrm{e})||p(\hat{z}_\mathrm{e})) = \int p(z_\mathrm{e}) \log \frac{p(z_\mathrm{e})}{p(\hat{z}_\mathrm{e})} dz_\mathrm{e} \tag{15}$$

$$= \int p(z_\mathrm{e}) \log p(z_\mathrm{e}) dz_\mathrm{e} - \int_{R_i} p(z_\mathrm{e}) \log p(\hat{z}_\mathrm{e}) dz_\mathrm{e} - \int_{R_o} p(z_\mathrm{e}) \log p(\hat{z}_\mathrm{e}) dz_\mathrm{e} \tag{16}$$

$$\ge \int p(z_\mathrm{e}) \log p(z_\mathrm{e}) dz_\mathrm{e} - \int_{R_i} p(z_\mathrm{e}) \log p(\hat{z}_\mathrm{e}) dz_\mathrm{e} - \int_{R_o} p(z_\mathrm{e}) dz_\mathrm{e} \cdot \log(\delta) \tag{17}$$

$$= O(\log \frac{1}{\delta}), \tag{18}$$

which is relatively large when $\delta \to 0$.

In an ideal scenario where the dataset is sufficiently large and evenly distributed, the region $R_o$ diminishes, effectively becoming negligible. Consequently, we could expect that:

$$KL(p(z_\mathrm{e}) \,||\, p(\hat{z}_\mathrm{e})) = O(1), \tag{19}$$

as $R_o \to 0$. Conversely, in our setting where fMRI samples are sparse ($n \ll d$), a substantial region of $R_o$ persists, indicating a significant divergence in the encoded latent space.

After vector quantization, the number of samples $n$ greatly exceeds the number of quantization bins $K$. Assuming there is no disproportionate concentration of probability mass within these bins, the KL divergence becomes:

$$KL(p(z_\mathrm{q}) \,||\, p(\hat{z}_\mathrm{q})) = \sum_{k=1}^{K} p(z_\mathrm{q}) \log \frac{p(z_\mathrm{q})}{p(\hat{z}_\mathrm{q})} = O(1). \tag{20}$$

As a result,

$$KL(p(z_\mathrm{q}) \,||\, p(\hat{z}_\mathrm{q})) \ll KL(p(z_\mathrm{e}) \,||\, p(\hat{z}_\mathrm{e})). \tag{21}$$

### A.5 Proof of Theorem 3.2

Assume that the high-dimensional latent space $A$ for $z_\mathrm{e}$ is confined within a closed hyperrectangle $[a_1, b_1] \times [a_2, b_2] \times \ldots \times [a_n, b_n]$ for each dimension. In a pretrained CLIP space as described in Radford et al. (2021), these bounds can be set to the extremal values obtained from encoding all pretraining images or texts.

Given any $\epsilon > 0$, one can choose a $\delta > 0$ such that $A$ is divided into a grid of smaller hyperrectangles. Specifically, we define a partition $(P_1, \ldots, P_d)$ where $P_i = (a_i = t_0 < t_1 < \ldots < t_{N_k} = b_i)$ with each interval $t_{j+1} - t_j$ being uniform and not exceeding $\delta$. Consequently, each subrectangle $S = [a'_1, b'_1] \times [a'_2, b'_2] \times \ldots \times [a'_d, b'_d]$ shares the similar volume $\Delta V_S$ and accommodates a integrated probability $\int_{S_j} P(z_\mathrm{e}) \, dz_\mathrm{e} = P(e_j)$.

Under the vector quantized encoder and for sufficiently small $\delta$, the quantized space can be further partitioned such that $P(e_k) = \sum_{j=1}^{J_k} P(e_{k_j})$, where $P(e_{k_j})$ represents the probability mass within the $j$-th partition of the $k$-th quantized space.

For each subrectangle $S = [a_1', b_1'] \times [a_2', b_2'] \times \cdots \times [a_d', b_d']$ of $P$ define its volume and bounds as:

$$v(S) = \prod_{i=1}^{d} (b_i' - a_i'), \tag{22}$$

$$m_S(f) = \inf f(x) : x \in S, \tag{23}$$

$$M_S(f) = \sup f(x) : x \in S. \tag{24}$$

Lower and Upper Riemann sums corresponding to the partition $P$ are then defined to be:

$$L(f, P) = \sum_{S \in P} m_S(f) \cdot v(S), \tag{25}$$

$$U(f, P) = \sum_{S \in P} M_S(f) \cdot v(S). \tag{26}$$

By the properties of Riemann integration, given any partition $P$ with norm $\|P\| < \delta$, it follows that:

$$U(f, P) - L(f, P) < \epsilon. \tag{27}$$

For each subrectangle $S$, we approximate the integrated probability over $S$ by selecting the 'average' value within this region, which is given by $\frac{P(e_k)}{\Delta V_S}$ and lies between $m_S(f)$ and $M_S(f)$.

$$L(f, P) \leq \int_S f(z_\mathrm{e}) dz_\mathrm{e} \leq U(f, P), \tag{28}$$

$$L(f, P) \leq \sum_{i_1=1}^{N_1} \cdots \sum_{i_n=1}^{N_n} \frac{P(e_k)}{\Delta V_S} \log \frac{P(e_k)}{\Delta V_S} * \Delta V_S \leq U(f, P). \tag{29}$$

Therefore, we have:

$$\sum_{i_1=1}^{N_1} \cdots \sum_{i_n=1}^{N_n} \left( P(e_k) \log \frac{P(e_k)}{\Delta V_S} - \epsilon \right) \leq \int_{z_\mathrm{e}} P(z_\mathrm{e}) \log P(z_\mathrm{e}) dz_\mathrm{e} \tag{30}$$

$$\int_{z_\mathrm{e}} P(z_\mathrm{e}) \log P(z_\mathrm{e}) dz_\mathrm{e} \leq \sum_{i_1=1}^{N_1} \cdots \sum_{i_n=1}^{N_n} \left( P(e_k) \log \frac{P(e_k)}{\Delta V_S} + \epsilon \right). \tag{31}$$

Consequently,

$$\lim_{\epsilon \to 0} H(z_\mathrm{e}, \epsilon) = -\sum_{i_1=1}^{N_1} \cdots \sum_{i_n=1}^{N_n} \left( P(e_k) \log \frac{P(e_k)}{\Delta V_S} \right). \tag{32}$$

As we consider the limit where $\epsilon \to 0$, it becomes feasible to represent the partitions of $A$ through their discrete counterparts.

We denote $H(z_e) = \lim_{\epsilon \to 0} H(z_e, \epsilon)$ as the entropy pf Riemann-Discrete approximated distribution of the embeddings after MLP $z_e = f_e(X)$ without vector quantization. Then, we have:

$$H(z_e) = -\sum_{k=1}^{K} \sum_{j=1}^{J_k} P(e_{k_j}) \log \frac{P(e_{k_j})}{\Delta V_S}. \tag{33}$$

$$H(z_q) = -\sum_{k=1}^{K} P(e_k) \log P(e_k) \tag{34}$$

$$= -\sum_{k=1}^{K} \sum_{j=1}^{J_k} P(e_{k_j}) \log P(e_k). \tag{35}$$

We operate under the hypothesis that the probability distribution is dispersed across the space, which precludes significant localization or the emergence of regions with disproportionately high probability mass. This is a plausible assumption within a space that has been pretrained with a large set of data, thereby approximating a well-spread distribution. Formally, we can express this as

$$J_k = O\left(\frac{L^d}{K \Delta V_S}\right), \text{ or to say } J_k = c_k \frac{L^d}{K \Delta V_S}. \tag{36}$$

where $c_k$ is a constant of order 1 ($c_k = O(1)$) and strictly positive ($c_k > 0$). In the case where the scale of the space $L$ is large and the dimensionality $d$ is much larger than the number of quantization bins $K$, the ratio $\frac{K}{L^d}$ becomes vanishingly small, implying that $c_k \ll \frac{K}{L^d}$, leading to the result:

$$P(e_k) = O\left(\left(\frac{L^d}{K \Delta V_S}\right) P(e_{k_j})\right), P(e_k) > \frac{P(e_{k_j})}{\Delta V_S}. \tag{37}$$

The implication here is that the entropy of the encoded space $H(z_e)$ is greater than that of the quantized space $H(z_q)$, accounting for the additional logarithmic factor:

$$H(z_e) - H(z_q) = O\left(\log\left(\frac{L^d}{K}\right)\right), H(z_e) > H(z_q). \tag{38}$$

The difference $\log\left(\frac{L^d}{K}\right)$ particularly large in our specified setting when the dimensionality $d$ is much less than the number of fMRI samples $n$, which in turn is substantially less than the number of quantization bins $K$, and considering the large size of the CLIP space denoted by $L$.

## A.6 FURTHER RESULTS ON FMRI INTERPRETATION

The visualization of voxel-wise importance maps of subject 2 and subject 3 is depicted in Figure 10 and Figure 11. Both figures illustrates that early layers of the video mapping show a focus on structural details of brain regions, while deeper layers and the VQ-fMRI encoder increasingly concentrate on abstract features. Foreground encoding exhibits significantly more activity compared to the background.

## A.7 FURTHER RESULTS ON 4D GENERATION

Additionally, figure 13 shows the overrall 4D effects where dynamic images rendered from different viewpoints at different timestamps. Figure 14 shows more samples with subjects 1-3.

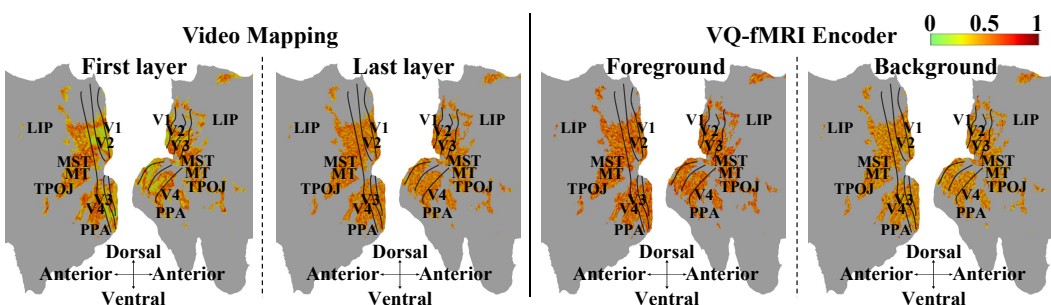

Figure 10: **Voxel-wise importance maps of subject 2.**

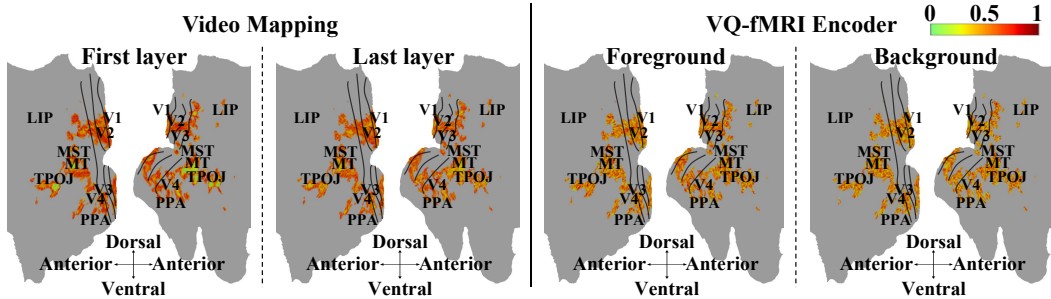

Figure 11: **Voxel-wise importance maps of subject 3.**

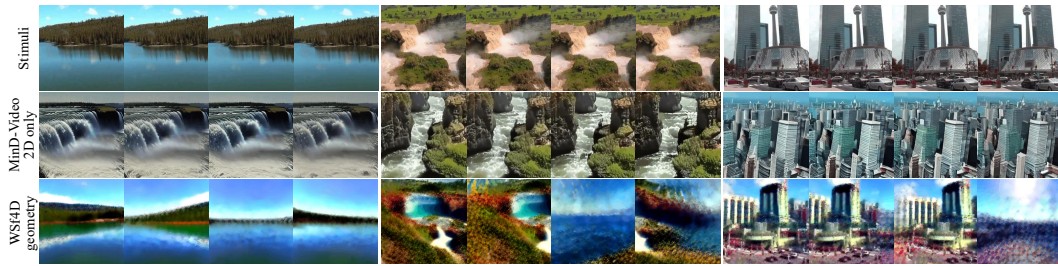

Figure 12: In background cases, WSf4D not only achieves consistent 360° rendering, but also delivers higher semantic accuracy with respect to ground truth stimulus.

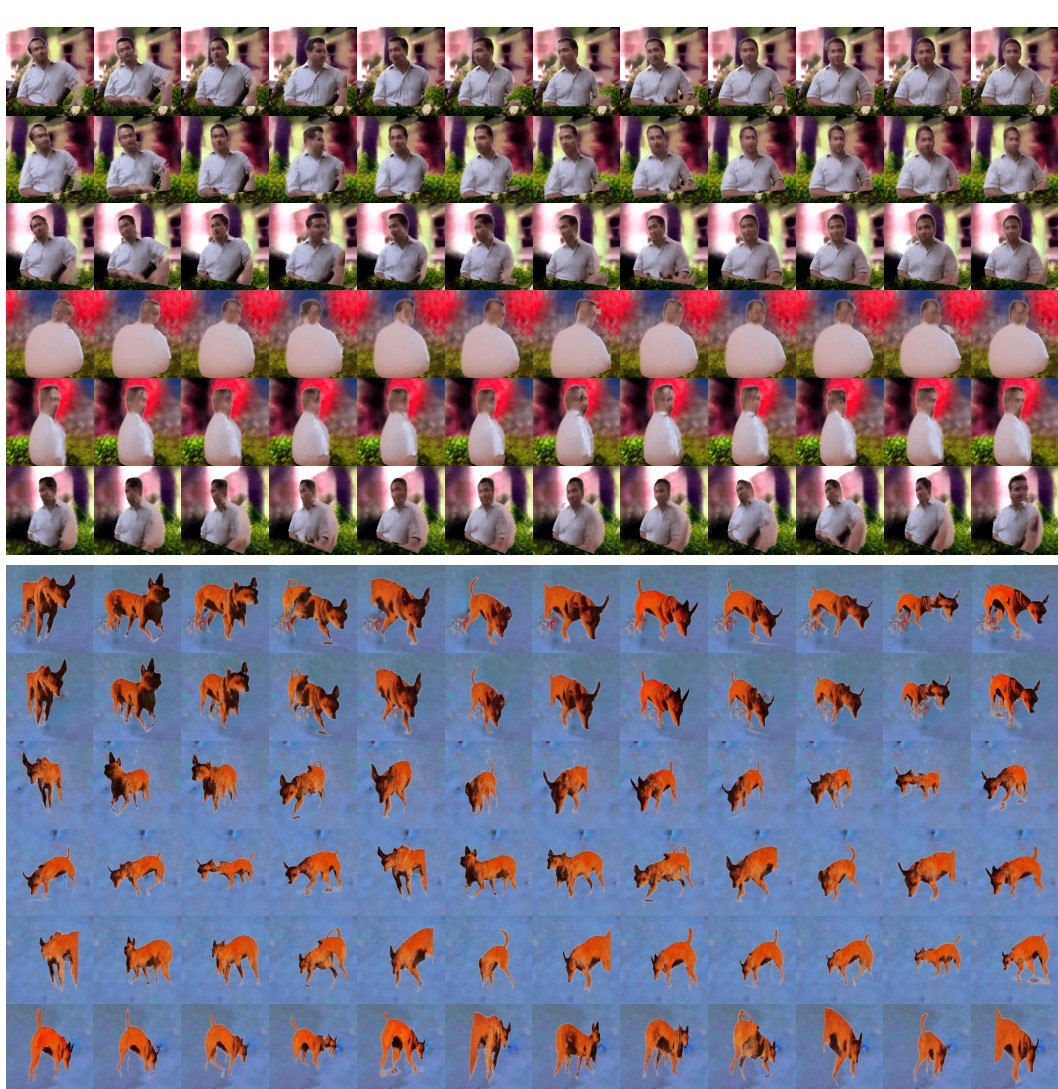

Figure 13: **4D results of two cases.** For each case, we show 6 viewpoints and 12 consecutive frames.

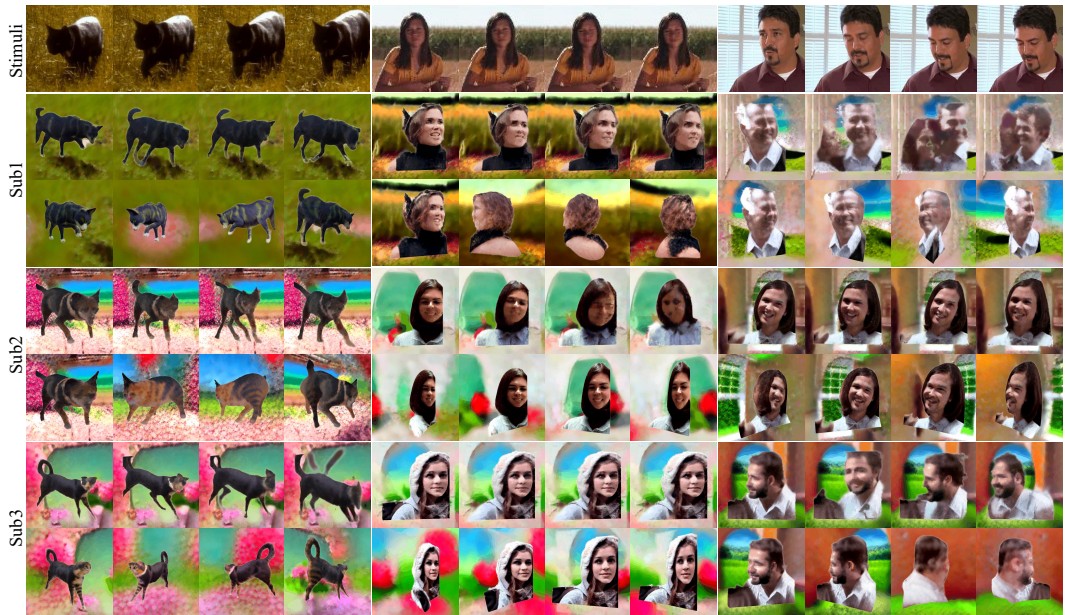

Figure 14: Samples from different subjects.

