# OpenReview forum: "Brain-to-4D: 4D Generation from fMRI"
_ICLR.cc/2025/Conference — ICLR 2025 Conference Withdrawn Submission_

### Official Review · Reviewer_A6t7 · 2024-10-28

**Soundness:** 2
**Presentation:** 2
**Contribution:** 3
**Rating:** 5
**Confidence:** 3

**Summary:**

The paper is well-structured and clearly introduces the Brain-to-4D model. The problem statement and the innovative approach of transforming fMRI signals into 4D scenes are described in great detail. The use of 2D video as weak supervision for 4D generation based on fMRI is innovative and intriguing. The paper presents an interesting method for decoding and reconstructing 4D scenes from brain signals, expanding the potential applications of BCIs in neuroscience and virtual reality. The use of a decomposition framework for scene generation is an exciting research direction that warrants further exploration.

**Strengths:**

1. Experiments in this paper appear to be rigorous and well-structured.
2. The proposal to use weak supervision for fMRI-based 4D scene generation is a unique contribution to the field of BCIs.
3. This paper effectively introduces new methods for decoding brain signals and reconstructing 4D scenes, which could have valuable applications in neuroscience and virtual reality.

**Weaknesses:**

1. The lack of comparison with other baseline models is a limitation. Since your model encodes and generates the foreground and background separately, it would be possible to compare the generation quality of the foreground and background independently rather than combining them in a single comparison. Moreover, having only Mind-Video (Chen et al., 2024) as a comparison seems insufficient; additional fMRI-to-video models, such as Mind-Animator (Lu et al., 2024), could be included. Although these models generate only 2D videos, comparing the images from the frontal view could provide a partial assessment of generation quality, as you have done in Fig. 4 and Table 1.


2. Most of the components in this method have already been explored in previous works, and this paper primarily integrates them to generate 4D representations. For example, the concept of VQ-fMRI (Chen, Qi, & Pan, 2023) is no longer novel, and the authors have subsequently extended its application in MindArtist (Chen et al., 2024). Another key component is the representation-to-4D generation part, where the paper also mentions in lines 890-891 that it uses the framework of DreamGaussian4D (Ren et al., 2023).


3. I noticed that you directly use the original image to supervise background generation (Fig. 2), which leads to parts of the foreground being mistakenly recognized as background, resulting in an impure background generation (Fig. 6, right). It may be beneficial to preprocess the supervision data to remove foreground interference, which could better leverage the advantages of foreground-background decoupling.

**Questions:**

1. What was the rationale for selecting VQ, and how does it specifically help in reducing noise in fMRI signals? You have shown an ablation study on VQ in Fig. 6 (left), but the categories overlap, and they do not correspond to the examples on image mapping in the right figure. More details would help clarify the method's scientific rigor.

2. How does the framework account for the temporal and spatial variations inherent in fMRI data? This is not explained clearly and requires more elaboration.

3. Could you provide more information on joint refinement? Specifically, details on the implementation of joint refinement, more examples, or an objective metric for comparison, as the effect shown in Fig. 7 doesn’t seem particularly significant.



References

[1] Chen Z, Qing J, Zhou J H. Cinematic mindscapes: High-quality video reconstruction from brain activity[J]. Advances in Neural Information Processing Systems, 2024, 36.

[2] Lahner B, Dwivedi K, Iamshchinina P, et al. Modeling short visual events through the BOLD moments video fMRI dataset and metadata[J]. Nature communications, 2024, 15(1): 6241.

[3] Chen, J., Qi, Y., & Pan, G. (2023, July). Rethinking visual reconstruction: experience-based content completion guided by visual cues. In Proceedings of the 40th International Conference on Machine Learning (pp. 4856-4866).

[4] Chen, J., Qi, Y., Wang, Y., & Pan, G. (2024). Mind Artist: Creating Artistic Snapshots with Human Thought. In Proceedings of the IEEE/CVF Conference on Computer Vision and Pattern Recognition (pp. 27207-27217).

[5] Lu, Y., Du, C., Wang, C., Zhu, X., Jiang, L., & He, H. (2024). Animate Your Thoughts: Decoupled Reconstruction of Dynamic Natural Vision from Slow Brain Activity. arXiv preprint arXiv:2405.03280.

[6] Ren, J., Pan, L., Tang, J., Zhang, C., Cao, A., Zeng, G., & Liu, Z. (2023). Dreamgaussian4d: Generative 4d gaussian splatting. arXiv preprint arXiv:2312.17142.

---

### Official Review · Reviewer_aXZR · 2024-10-29

**Soundness:** 2
**Presentation:** 1
**Contribution:** 2
**Rating:** 3
**Confidence:** 5

**Summary:**

This manuscript aims to enhance brain-computer interfaces by generating immersive 4D visuals, including videos and 3D elements, directly from fMRI signals. To tackle the challenges of training data collection and the variability of brain fMRI data, an approach called WSf4D utilizes weak supervision through foreground-background decomposition and multifaceted vector quantization for noise reduction. Extensive experiments demonstrate that WSf4D effectively generates multi-view 4D scenes that are semantically aligned with raw brain signals, showcasing significant advancements in neuroscience applications.

**Strengths:**

1.The authors conducted extensive and comprehensive experiments to validate the effectiveness of their proposed model.

2.The authors proposed the use of vector quantization to achieve fMRI decoding, which mitigates the impact of noise in fMRI data and enhances decoding performance. Additionally, the authors provided theoretical evidence demonstrating the superiority of employing vector quantization.

**Weaknesses:**

1. Writing issues.

The authors exhibit significant issues in writing and expression, which considerably hinder the reading experience and further comprehension of their intentions.

（1）The authors did not provide a detailed explanation of the architecture shown in Figure 2, which makes it difficult to understand the model. The authors are requested to separately describe the training and inference processes of WSf4D. Additionally, they should clarify the origin of the “Fg” in Figure 2(a)—whether it was pre-trained by themselves or derived from an existing pre-trained model. Lastly, the authors are asked to explain the meaning of the dashed line above “Supervision” in Figure 2(a).

(2) From Section 4.2, it is evident that both Foreground 3D-aware diffusion and Background 3D-aware diffusion are pre-trained models from other works and are not contributions of this paper. Therefore, it raises the question of why the authors dedicate such extensive space from lines 239 to 269 to describe their loss function.

2. Experiment issues:
（1）The authors have included too few baselines for comparison in Table 1, and under the front view setting, WSf4D does not demonstrate significant superiority over MinD-Video.

（2）The ablation experiments presented in this paper consist only of qualitative comparisons and lack quantitative analysis, which may hinder the validation of the effectiveness of the proposed components. For instance, in Figure 7, it is challenging to discern the improvements brought by the refinement stage through visual inspection alone. The authors are requested to include quantitative ablation experiments.

**Questions:**

1.In the dataset used by the authors, the participants have not actually viewed the 4D content; however, the authors generated 4D content from fMRI data. The authors are requested to explain the rationale and necessity of this approach.

2.The characters in the second and third rows of Figure 4 appear to be identical; however, these results are derived from two different models. The authors are requested to provide an explanation for this observation.

3.What is referred to as "video mapping layers" in line 504 has not been mentioned in the preceding text. The authors are requested to provide an explanation for this concept.

---

### Official Review · Reviewer_T2QY · 2024-10-31

**Soundness:** 3
**Presentation:** 2
**Contribution:** 2
**Rating:** 5
**Confidence:** 5

**Summary:**

The paper introduces Brain-to-4D, which aims to generate 4D visuals (including both video and 3D) directly from brain fMRI signals. The authors propose a Weakly Supervised decomposed fMRI-to-4D generation approach, named WSf4D, which addresses the challenges of acquiring brain signals for multi-view 4D stimuli and the large variation in brain fMRI data. The method involves foreground-background decomposition for supervision and fMRI multifaceted vector quantization for noise suppression. The paper demonstrates the application of WSf4D in neuroscience and diagnosis by encoding distinct visual cortex groups and shows that it can generate multi-view consistent 4D scenes that are semantically aligned with raw brain signals.

**Strengths:**

The paper presents an approach in BCI technology by attempting to generate 4D content from fMRI signals, which is a novel and creative extension of current BCI capabilities. The idea of decomposing the scene into foreground and background for weakly supervised learning is innovative.

The methodology, including the WSf4D framework and the use of vector quantization for fMRI signals, appears to be well-thought-out and technically sound.

The paper is well-structured and clear in its presentation of the problem, the proposed solution, and the experimental results.

**Weaknesses:**

1. Although the authors propose a new decoding method for brain-to-4D, the experimental validation appears to be weak. First, the fMRI dataset used in the experiments consists of 2D videos watched by the subjects, which do not effectively stimulate the brain's 3D representation information. Why did the authors choose to use fMRI evoked by 2D stimuli to reconstruct 3D information? Furthermore, since there is no ground truth for the 3D information, how can the quality of the 3D reconstruction be effectively evaluated?

2. From the reconstruction results in Figures 4 and 7, the reconstruction effect of WSf4D does not seem promising. For instance, in Figure 7, the color, posture, and size of the dog are poorly reconstructed. Under such circumstances, what is the significance of pursuing consistency across multiple viewpoints? Even if a high degree of consistency is presented across viewpoints, is this information decoded from the brain signals or is it prior information from the diffusion model?

3. The authors only used one fMRI dataset and only compared it with one method, MinD-Video, which is insufficient. In the field of video reconstruction, there are already multiple datasets and reconstruction methods available for comparison.

4. Since this paper is about 4D reconstruction, it is difficult to appraise the temporal coherence and multi-view consistency solely through image visualization. Is there an anonymous project homepage or link to display the relevant experimental results?

**Questions:**

see above.

---

### Official Review · Reviewer_z9T2 · 2024-11-04

**Soundness:** 3
**Presentation:** 2
**Contribution:** 3
**Rating:** 5
**Confidence:** 4

**Summary:**

In this paper, the authors propose a new task, termed Brain-to-4D, which aims to translate fMRI signals into 4D visual representations, including video and 3D structures. Due to practical limitations in acquiring brain signals for multi-view 4D data, the authors introduce WSf4D, a novel weakly supervised technique that utilizes foreground-background decomposition and multifaceted vector quantization to enhance fMRI-to-4D generation. The key idea is to leverage partial supervision to establish correspondences between two modalities: 4D object targets, representing the foreground, and a 3D background in video format.

**Strengths:**

1 - The transformation of fMRI input into distinct foreground and background representations, followed by recombination into a cohesive 4D visual format, is a novel approach.

2 - The methodology is well-defined with solid mathematical foundations.

3 - Comprehensive ablation studies demonstrate the effectiveness of the proposed architecture.

**Weaknesses:**

1 - The problem formulation lacks clarity, and the flow of the abstract could be improved.

2 - To clarify how the VQ-fMRI encoders address the challenge of distinguishing meaningful brain dynamics from noise, could the authors provide more details on the design or training methods used to differentiate signal from noise in the fMRI data? Additionally, it would be helpful for the authors to discuss any limitations or assumptions related to noise suppression in their approach.

3 - Could the authors elaborate on how temporal information is represented and processed within the model architecture? Specifically, it would be useful to clarify whether the model assigns a latent vector to each fMRI time point, and to explain the significance of $K$ in the equation. Does $K$ represent the number of time points in $g_{Fg} \in \mathbb{R}^{K_{Fg} \times D_{Fg}}, \quad g_{Bg} \in \mathbb{R}^{K_{Bg} \times D_{Bg}}$ ? This additional information would help assess the model's suitability for capturing temporal dynamics in fMRI data.

4 - Could the authors discuss the potential limitations of mapping fMRI data into a text embedding $Z$ and how they address any associated information loss? It would also be helpful to explain how their approach compares to a more direct mapping from fMRI data to the target, if feasible, and the potential impact on model performance.

5 - To facilitate evaluation, could the authors include experimental comparisons with relevant baseline methods, if available? For computational complexity, it would be useful if the authors could provide details on runtime and hardware requirements, and discuss how their pipeline compares to other methods in terms of computational efficiency as well.

6 - The paper requires revision to address minor typographical errors, such as changing "use experience" to "user experience" in the introduction and correcting "an shared" to "a shared" in the methods section.

**Questions:**

1 - I would like to seek clarification regarding the introduction of the new task labeled Brain-to-4D in the context of existing research on "decoding visual stimuli" within computational neuroscience, computer vision, and machine/deep learning. It appears that this task may be a spatiotemporal extension of conventional visual stimulus decoding techniques. Could the authors elaborate on the motivation behind defining Brain-to-4D as a distinct task?

2 - Is there any possibility to check other metrics like Visual information fidelity (VIF) along with reported SSIM?

---

### Note · Authors · 2024-11-13

I have read and agree with the venue's withdrawal policy on behalf of myself and my co-authors.